# Role of active patient involvement in undergraduate medical education: a systematic review

Stijntje Willemijn Dijk  ,[1] Edwin Johan Duijzer,[2] Matthias Wienold[3]

SWD and EJD are joint first authors.

[1]Erasmus University Medical Center, Rotterdam, Zuid-Holland, Netherlands
[2]University Medical Centre Groningen, Groningen, Netherlands
[3]Wissenschaftliche Dienste und Projektberatung, Berlin, Germany

**Correspondence to**
Stijntje Willemijn Dijk;
340974sd@student.eur.nl

## ABSTRACT

**Objectives** To identify the scope of active patient involvement in medical education, addressing the current knowledge gaps relating to rationale and motivation for involvement, recruitment and preparation, roles, learning outcomes and key procedural contributors.

**Methods** The authors performed a systematic search of the PubMed database of publications between 2003 and 2018. Original studies in which patients take on active roles in the development, delivery or evaluation of undergraduate medical education and written in English were eligible for inclusion. Included studies' references were searched for additional articles. Quality of papers was assessed using the Mixed Methods Appraisal Tool.

**Results** 49 articles were included in the review. Drivers for patient involvement included policy requirements and patients' own motivations to contribute to society and learning. Patients were engaged in a variety of educational settings in and outside of the hospital. The vast majority of studies describe patients taking on the role of a patient teacher and formative assessor. More recent studies suggest that patients are increasingly involved in course and curriculum development, student selection and summative assessment. The new body of empirical evidence shows the wide range of learning objectives was pursued through patient participation, including competencies as professional, communicator, collaborator, leader and health advocate, but not scholar. Measures to support sustainable patient involvement included longitudinal institutional incorporation, patient recruitment and/or training, resource support and clear commitment by faculty. The importance and advantages of patient involvement were highlighted by students, faculty and patients themselves; however, organisations must continue to consider, monitor and take steps to mitigate any potential harms to patients and students.

**Discussion** This systematic review provides new knowledge and practical insights to physicians and faculty on how to incorporate active patient involvement in their institutions and daily practice, and provides suggested action points to patient organisations wishing to engage in medical education.

## INTRODUCTION AND RATIONALE

In recent decades, the involvement of patients in medical education has been advocated for increasingly and has become common practice adopted by reformers of medical education.[1] Patients and their narratives are no longer just used as subjects for 'learning material' in clinical training. Towle *et al* identified different levels of involvement, from paper-based involvement to involvement at the institutional level as codesigners of the medical curriculum in addition to sustained involvement as patient teachers in education, evaluation and curriculum development.[1–4] Medical educators are now seeing the value of linking medical students with patients and their families and communities to foster awareness of the importance of longitudinal relationships, to improve students' social interaction skills and to facilitate learning of coping with illness in the real world.[5 6]

Despite an increasingly collaborative role of patients in medical education, there is much to be learnt about how to embed it, and how to develop systematic, institution-wide approaches to planning patient involvement in all levels of medical education.[3 7] The drive towards a more equal partnership in clinical decision-making and patient-centred care, fuelled by national and international guidelines, promotes the expansion of the

### Strengths and limitations of this study

► This systematic review is the first of its kind focused specifically on undergraduate medical education, providing practical guidance to educators, students and patients with ambition to improve work in healthcare professionals' education.

► The study provides novel insights in the wide range of learning objectives pursued through patient participation, the educational settings and roles in which patients participate and practical support systems that enable patient engagement.

► As many articles written by patients on their experiences in involvement in medical education may only be found in grey literature, including blog posts, conference statements and patient organisation newsletters, this review may have missed their viewpoints.

efforts towards developing a culture where partnership in medical education becomes the norm.[3 5–15]

As researchers have used varying definitions of active patient involvement in medical education, they have used varying inclusion criteria in their literature searches. This has resulted in overlap of included papers, and limited the generation of a common theoretical framework and terminology.[3 16] Previous studies have identified major gaps in the knowledge base relating to short and long-term learning outcomes, ethical issues, psychological impact and key procedural contributors like recruitment, selection and preparation. There is also limited information of the cost-effectiveness of active patient involvement.

Since the publication of the last systematic reviews[1 17–19] and non-systematic reviews[2 8 20] of patient involvement in medical education, many new studies have been published. Previous reviews addressed only the patient teacher role,[1] teaching and assessing one specific skill (intimate examination),[17] included simulated patients,[17] included postgraduate medical education[18] or addressed all healthcare professions.[2 17 20] A recent systematic review provided a comprehensive overview of the involvement, outcome and reason behind involvement mainly from learner's perspective.[19]

Our paper reviews and summarises the most recent literature using a broad definition of patient involvement consisting of any form of involvement that is beyond merely incidental passive involvement, in any field or setting of undergraduate medical education. By adopting this definition we are able to extend the scope and amount of research data in order to increase the practical knowledge base on active patient involvement and in order to give ground to an improved theoretical framework and common terminology. Our study takes a novel approach by focusing primarily on the patients' perspective on their involvement.

## METHODS
### Design
Our literature search employed a systematic review method looking for active patient involvement in medical education defined as the direct involvement of real patients and community members in the development, delivery or evaluation of undergraduate education of medical students.

### Search strategy
The authors performed a search through PubMed on 12 July 2018. The search terms used on their own and in combination included: *patient\**, *communit\**, *involvement*, *engag\**, *cooperat\**, *collaborat\**, *represent\**, *medical education*, *curriculum*, *medical student\**. Search criteria were reviewed by a hospital information specialist. The full search strategy can be found in online supplementary file 1.

All articles published in English between 1 January 2003 and 12 July 2018 and reporting primary empirical research that addressed the active participation of patients

in undergraduate medical education were eligible for inclusion. Studies with simulated patients or actors, patients solely undergoing examinations or patients who were only being observed in wards were excluded from the review.

We assessed articles based on title and abstract in the first round, and in a second round based on full text. References of all included articles were analysed for additional studies that matched the original inclusion criteria. All reviews that complied with the inclusion criteria were additionally assessed for relevant references. Only original research articles were included in the final analysis.

### Quality assessment of included studies
As our review included papers of qualitative, quantitative and mixed methods designs, two authors (SWD and ED) applied the Mixed Methods Appraisal Tool to assess the methodological quality of studies (online supplementary file 2).[21] Studies were not excluded based on assessed methodological quality.

### Data extraction and synthesis
All authors (SWD, ED, MW) used a prepiloted standardised form to extract data from included studies. A second author checked if the extracted data were accurate, and discrepancies were resolved through discussion. The following data were recorded: authors, year of publication, journal, country intervention, study type, abstract, study setting, financial aspects reported, number of patients in intervention, number of students in intervention, patient characteristics, patient motivations to join, recruitment practices, training and preparation practices, role of patient organisations, type of patient involvement, outcome measurement, organisational remarks on sustainability and pursued learning outcomes. To explore this range of learning outcomes, we categorised intended learning outcomes according to the CanMEDS framework as a commonly applied competency framework within medical schools.[22] We organised extracted data in related themes to explore connections and discrepancies between data elements. We opted not to use any of the existing frameworks for grouping potential roles patients take on. In the Discussion section, we compare our findings of the diversity of roles with the existing taxonomy by Towle *et al*.[2]

### Patient and public involvement statement
The initial impulse for this study initiative followed a collaboration between the authors as members of the International Alliance of Patients' Organizations (IAPO) and the International Federation of Medical Students' Associations (IFMSA). MW, patient representative and co-author, was involved as an equal partner in all stages of the research project including project initiation, study design, data analysis, discussion and writing of the paper. The initial draft of this paper was presented and discussed at the European Patient Forum 2019 in a plenary session

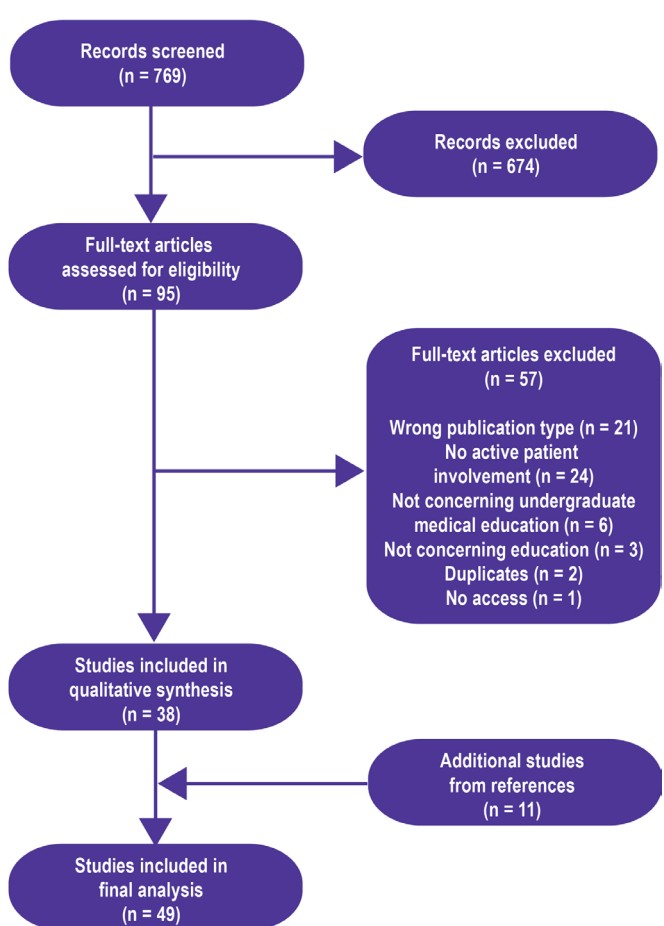

**Figure 1** Results of the systematic literature search.

with 300 patient representatives present, feedback from which has been incorporated into the final paper.

## RESULTS
### Study selection

The initial search resulted in 769 articles, of which 95 were selected for further review based on the title and/or abstract. These 95 articles were independently read by two authors (SWD and ED) and included based on the specified criteria. Consensus between reviewers was 91.2%. The remaining articles were included based on consensus after a short discussion. Main reasons for exclusion where wrong article type (conference abstracts or commentaries) and studies that did not concern active participation of patients. The review of references resulted in 11 additional articles for inclusion. The characteristics of the 49 studies that met inclusion criteria are presented in figure 1 and online supplementary file 3.

We used the extracted data from included studies to synthesise the evidence in the following subsections:

► Rationale for involving patients in medical education.
► Patient recruitment and selection.

► Patients' preparation to participate in medical education.
► Roles patients take on in medical education.
► Learning objectives pursued through patient involvement.
► Concerns about the involvement of patients.
► Patients' views on the impact of their involvement.
► Financial implications of patient involvement.
► Roles of patient organisations.
► Measures to ensure the sustainability of patient involvement.

### Rationale for involving patients in medical education

Several authors referred to government policy mandating patient participation in medical education, namely the UK Department of Health and the UK General Medical Council,[16 23 24] the Australian Medical Council[25] and the WHO[5] as a rationale for their patient involvement initiatives. Besides these political drivers, cited rationales were: to teach students patient-centred and interprofessional care[24 26–28]; to introduce students to chronic illness care[29–31]; to create a multicultural learning environment; to practise social accountability and an inclusion agenda[32–37]; to make education more engaging, powerful and transformative[38]; and to empower patients.[39] Patients mentioned that they felt a sense of responsibility to the broader community in shaping the future health workforce[24 32] and improving the healthcare system.[24 33 40]

### Patient recruitment and selection

The most frequently reported methods to recruit patients were through existing university partnerships and existing programmes,[28 33 41–46] advertisements through press or social media or posted in health facilities,[26 34–36 47 48] through community and patient organisations, through personal connections and previous participants,[32 38 49–51] and through health professionals.[52–55] In one project, where students shadowed a patient with a chronic condition, students were asked to recruit patients themselves.[30]

Selection criteria were generally set up broadly, inviting any patient or community member. General criteria for patient educators included good communication skills, affinity for teaching, aptitude for further learning, enthusiasm, time to commit to the study, as well as being fully mobile and being able to cope with repeated physical examinations.[48 54 56] In some cases patients were required to have representative physical signs of their disease.[53 54 56 57]

### Patients' preparations to participate in medical education

Twenty articles mention preparation of patient teachers. The duration of the preparation ranged from substantial training sessions of 100 hours in total[42] to the majority of programmes providing solely written information or a single orientation session of 1–1.5 hours.[24 30 32 39 43 49 52 57] Training programmes for patients in teaching musculoskeletal (MSK) skills were the most extensive and were delivered by medical or educational experts.[42 47 54 56 58 59]

Other preparatory sessions were less formal and were facilitated by faculty educational experts, students or peers.

The primary aims of the preparatory programmes varied. One study underscored the importance of patient educators being aware of the course goals in order to safeguard student learning outcomes.[53] Others mentioned aiming to serve the needs of patients in building their confidence, providing skills training[57] and providing knowledge related to the educational process.[23 39 48 60] These sessions addressed approaches such as problem-based learning, how to deliver a presentation, cofacilitation methods and how to provide effective feedback. Preparation also provided opportunities for anticipating benefits and challenges such as conflicts, emotions, unmet expectations, using methods of coaching, supervision and debriefing.[40 61]

One article mentions finishing the preparation of patient educators with a short quiz as an assessment tool and having a graduation session before starting to teach.[28] Another article describes the use of a post-training satisfaction questionnaire to help ensure that patient educators felt ready to teach.[47] The majority of papers did not address assessing patients prior to them taking on their roles.

Towle *et al* discussed the tension between preparedness of patient educators and authenticity of education in both form and content.[49] They highlight the critical role of the community organisation representatives who can be brokers between the two cultures of academia and community. One article describes an intervention in which the patient educators explicitly have not received training, so the student–patient encounters would be as authentic as possible.[53] Another article described that the collaboration between patients and educators allows for mutual learning without an authentic patient perspective being lost.[39]

## Roles patients take on in medical education

The main categories of roles that we identified are divided in the areas of a patient as a teacher, an assessor, a curriculum developer and a student selector (table 1).

### Patient as teacher

The role of a teacher was cited most frequently. Patient teachers gave clinical skills practicals on history taking and physical examination sessions on their own condition such as MSK disorders.[42 47 54 56 58 59 62] They were trained to teach students skills and deliver immediate feedback, which stimulated further learning.[56]

Several groups of patients with disabilities acting as patient teachers with disabilities gave practicals on communication skills and history taking.[48 53 57] People living with HIV participated as teachers during a simulated clinical encounter in which students provided counselling.[60]

In addition to clinical skills, patients taught students about their experiences of overall management of care, and the personal aspects of their lives. These ranged from practical physical and home adjustments, to psychological, social and behavioural issues impacting them and their family. Teaching was done through panel discussions and small group sessions[24 43 63] as well as visits to the community and patients' homes.[31 52] Patient teachers with chronic conditions acted as mentors, and met regularly with students.[27 29] Patients taught students patient-centredness and interprofessionalism,[23 24 31 38 49] community-centredness, cultural

| Role | Specification |
| --- | --- |
| Patient teacher | Deliver clinical skills sessions on history taking, counselling and physical examination. Deliver formative feedback during teaching sessions. Share experiences in healthcare or personal aspects of their lives in teaching sessions, small group sessions, individual mentorship and coaching, or through the creation of videos. |
| Patient assessor | Deliver formative feedback during teaching sessions. Perform summative assessments during OSCEs. |
| Patient curriculum developer | Evaluate the medical programme. Act as a curriculum steering committee member. Were consulted to provide recommendations on disease-specific courses through focus groups, world cafés or discussions with community leaders. Participate in development of courses related to their illness or social conditions, or overarching competencies and community-based learning. Develop courses delivered by patient teachers autonomously. Identify end competencies for graduates. Identify desired curriculum characteristics. Be consulted on the strategic development of a new medical school department. |
| Patient selection committee member | Participate in the selection of students applying for the medical programme. Assess candidates' communication skills, sensitivity, compassion and empathy towards societal contexts and needs. |

**Table 1** Identified patient roles in medical education

OSCE, objective structured clinical examination.

competence and ethics.[33 64] Patients could choose their own teaching method, such as telling their stories and stimulating reflection.[38] A group of patients living with intellectual and developmental disabilities also participated in the creation of learning materials, through videos sharing their perspectives and stories.[65]

Most patients in the study by Jackson *et al* considered themselves not as teaching, but having a role of partnership, explanation and sharing certain aspects of their illness.[52]

### Patient as assessor of students' competence

In addition to formative student assessment, such as feedback during teaching sessions, patients participated in high stakes summative assessments, such as the final year objective structured clinical examination (OSCE).[39] Patients also provided written feedback to student essays, which were used for formal assessment.[39] Patients assessed mostly non-cognitive domains of student performance.[35 46] Medical educators interviewed by Jha *et al* believed there was a role for patients in assessing whether students made them feel at ease and whether students asked the right questions.[66] While patients and medical educators in the study by Raj *et al*[54] praised patients' assessments, students expressed their concerns whether patients could reliably assess clinical skills or whether they were likely to be too lenient.[54]

### Patient role in curriculum development

Community members were motivated to participate in curriculum development.[35] While they were not seen as medical experts, they did have an interest in ensuring optimal healthcare for themselves and their families.[26]

Community members actively participated in the planning, implementation and evaluation of the educational programme.[46 49] Patient teachers had autonomy from the stage of planning to the stage of delivery of teaching.[43 67] Several patients were members of the steering committee for the psychiatry curriculum[28] and the interprofessional education curriculum.[49]

Patients were involved in the development of courses related to their illness or social conditions.[35] Aboriginal delegates provided recommendations for the development of an Aboriginal health curriculum and community placement.[33 44] Focus group meetings with Native Hawaiians were held to define a cultural competencies and health disparities curriculum.[37] A world café discussion was similarly held to inform the curriculum on transgender health.[36] Minority community members provided input on curricular design, especially on the content of the cancer disparities curriculum.[35]

Beyond the disease or competency-specific courses, patients were involved in consultative meetings with stakeholders to identify desirable attributes, competencies of graduates and development of a community-based learning environment.[45] Patients were also consulted on the desired characteristics of the curriculum.[68] One medical school sought input for the strategic development

of the department of population health in a new medical school through focus groups.[69]

### Patient role in selection of students to medical schools

Community members were invited to join a panel together with clinicians and academic staff members to select students applying for the Graduate Entry Medical Program.[32] Members of the community were invited to be a part of the student selection process and team, especially in assessing candidates' communication skills as well as sensitivity, compassion and empathy towards social contexts and societal needs.[45]

### Collaboration between faculty and patients

The role of faculty members in the collaboration with patient teachers varied. Some patient teachers worked under the supervision of a clinical preceptor.[60] In other sessions, patients were cofacilitators with practitioners.[23] Workshops were led by patient teachers and facilitated, but not controlled, by faculty. The faculty member's role was to support the direct learning between students and mentors.[49] Faculty was not always present in meetings but could provide background support, such as setting broad topics for discussions.[46] Patient teachers stated that programme support was essential for participation, allowing them to transform from teaching individual messages to teaching universal lessons.[43]

Lay participants of one study regarded sharing of curriculum ownership as necessary to acknowledge the importance of lay perspectives, whereas faculty participants presumed ownership of curriculum development.[26] Faculty in the study by Jha *et al* were not clear on how to involve patients more fully in assessments or course development, nor were they convinced of the appropriateness of doing so. Some faculty members expressed their experiences of working with patient assessors and course developers as tokenistic.[66]

### Learning objectives pursued through active patient involvement

Learning outcomes of patient participation were quantitatively assessed on the subject of MSK examination skills in four randomised controlled experiments[42 47 54 62] and two further studies.[56 59] No difference was observed in increased structured clinical examination (OSCE) scores when comparing sessions delivered by trained patient educators with sessions delivered by rheumatology staff together with a passive patient undergoing examination[54 62] and sessions with a non-MSK specialist physician.[42] In the experiment by Humphrey-Murto *et al*, significantly fewer faculty-taught students failed (0 out of 32) than patient educator-taught students (5 out of 30).[62] Students rated faculty educators higher than patient educators (4.13 vs 3.58 on a 5-point Likert scale).[62]

When students were taught by a patient teacher in addition to the regular faculty-led sessions, their OSCE scores increased more compared with students participating in the regular curriculum.[47] An intervention by de Boer *et*

at[59] offered students the opportunity to participate in two non-obligatory real patient learning sessions in the preclinical MSK disorders block.[59] Students who participated scored significantly higher at the end-of-block test.

Oswald *et al* examined how teaching was different between patient educators and physician educators when teaching MSK physical examination skills.[58] Video recordings show that trained patient educators were more consistent in content and style by consistently covering all major joints. Bokken *et al*[53] assessed student's perspectives on instructiveness of real patients versus simulated patients.[53] Overall instructiveness was marked high. Students regarded real patients as more authentic and the encounters more useful in practising physical examination.

In the intervention study by Jaworksy *et al*, medical students provided HIV test counselling to patient instructors.[60] Preintervention and postintervention scores of the validated Health Care Provider HIV/AIDS Stigma Scale[70] demonstrated a significant decrease (68.74 vs 61.81). Students reported increased comfort in providing HIV-related care (10.24 vs 18.06). Similarly, students in intervention studies with patient teachers living with physical or mental disabilities demonstrated an improved attitude,[28] increased levels of comfort in communication,[57] increased levels of self-efficacy and confidence,[63 65] and higher mean performance scores across all interview stations when compared with a control group.[65]

Students in the study by Rees *et al* described the encounters with patients as more motivating compared with textbook learning.[71]

Wide ranges of learning outcomes of education with patient participation were mentioned in the qualitative studies identified by this review. To explore this range of outcomes a categorisation is used according to the CanMEDS framework, developed by the Royal College of Physicians and Surgeons of Canada[22] (table 2).

### Communicator

Several authors mentioned patient-centred care as the main outcome of education involving patients.[23 27 29 49 67] Patient-centredness included the ability to see patient mentors as individuals,[27] the importance of patient autonomy and expertise in care,[64] adopting a non-patronising and non-judgemental attitude,[55] recognising patients' needs[41] and seeing the patient as a capable part

| Relevant CanMEDS role | Specification of learning outcomes in reviewed studies |
|---|---|
| **Table 2** | Aspired learning objectives for medical students based on the CanMEDS framework |
| Communicator | Apply a patient-centred approach to interviewing and care. Adopt to the unique needs and preferences of each patient as an individual, recognising their needs. Communicate using a patient-centred approach that encourages patient trust and autonomy, recognising their expertise in care and seeing them as part of a team. Create an environment for patient comfort, dignity, privacy, engagement and safety by using a non-patronising and non-judgemental attitude, recognising biases. Apply communication skills to share information and explanations that are clear and accurate, checking for understanding, using communication skills that help patients make informed decisions. |
| Collaborator | Work effectively with physicians and colleagues in the healthcare professions through interdisciplinary teams. |
| Leader | Contribute to the improvement of healthcare delivery through understanding the broader healthcare system, and how it affects patients. |
| Professional | Demonstrate a commitment to patients and society by applying best practices and adhering to high ethical standards, dealing with ethical complexity of clinical practice. Demonstrating a commitment to the profession, reflecting on role models and professional identity, keeping fellow physicians and oneself to high professional standards and understanding patient views on clinical errors. Demonstrating commitment to physician health and well-being by learning to cope with uncertainties, emotions and stress. Exhibit appropriate professional behaviours and relationships, demonstrating respect for diversity, respect and compassion. |
| Health advocate | Responding to individual patient's health needs by advocating with the patient within and beyond the clinical environment, specifically for patients in vulnerable situations. Awareness of the importance of physician and patient advocacy. Working with communities or patients to identify determinants of health that affect them. |
| Scholar | No paper explicitly described the aim of developing the role of scholar. |
| Medical expert | Performing patient-centred clinical assessments and establishing a management plan. Establishing plans for ongoing continuity of care. Understanding the complexity of practising medicine. Integration of theory into practice. |

of the team.[41] Jha *et al* pointed out that active patient involvement by itself demonstrates an equal partnership[66] and Rees *et al* concluded that this approach helps students to develop a holistic perspective of healthcare.[71]

More generally, patient participation was associated with increased understanding of the importance of communication,[27 29] building and improving communication skills,[55 71] empathy, listening skills and respect.[71]

## Collaborator
McKinlay *et al* described an education programme in which students undertake a home visit to a patient with a chronic condition,[31] where students demonstrated increased understanding of interdisciplinary teams in management of chronic conditions in their reflective assays. Four authors described interprofessional education programmes in which patients are involved.[23 27 44 49]

## Leader
In a longitudinal mentor programme with medical, physical therapy, occupational therapy, nursing and pharmacy students teaming up with a patient mentor students reported a deeper understanding of the healthcare system.[27] A yearlong student mentor programme gave students an experience in and appreciation of continuity of care.[55]

## Professional
Various qualitative studies suggested that patient involvement can attribute to dealing with ethical complexity in clinical practice and patients' perspectives on clinician error[64] and developing reflective skills.[29 48 55] Reflecting on role models some authors referred to broadening understanding of the role of the healthcare provider,[27] qualities of remarkable clinicians that inform personal ideals,[64] creating a future professional model[55] and professional identity.[71] Experiences with real patient educator encounters could also help in coping with uncertainties, emotions and stress.[71]

Exposure to patient educators from within specific patient or minority groups helped students increase positive attitude towards chronic conditions and elderly,[27 31] patients with mental health problems[29] or disabilities.[48]

## Health advocate
Students reflected on the importance of patient advocacy in day-to-day practice in a study on experiences within an ethics and professionalism module with patient mentors.[64] More specifically, students were empowered to advocate for patients when they are in vulnerable situations. One of the aims of the education programme described by Saketkoo *et al* was to develop an awareness of the impact of physician advocacy, specifically in the context of people with disabilities.[63] A pretest and post-test showed that this awareness increased significantly with the programme.

## Scholar
No programmes have explicitly described the aim of developing the competency of scholar.

## Medical expert
The role of medical expert integrates all other roles by applying medical knowledge, clinical skills and professional values in the provision of high-quality and safe patient-centred care. Two articles mentioned that patient participation supports students' learning by recognising the complexity of practising medicine.[41 48] Jha *et al* described the patient as providing an illustration of the theory in practice, thus enhancing students' understanding and recall.[66]

## Concerns about the involvement of patients
Various authors have also raised concerns when involving patients as teachers in medical education. Some faculty educators were concerned that patient stories might be so traumatic that students would require support or debriefings to deal with the resulting emotions.[66 71] In a qualitative study, students felt 'pressured' by service users asking them for information and advice, rather than asking their clinicians, or when service users divulged information to students that they had not told their clinicians.[71] Students worried about giving incorrect information to patients.

Students expressed reservations that they were only getting the view of one person, which could lead to a biased perspective.[23] Students were also concerned that patients might have difficulty discriminating between poor and good performance, and are likely to be too lenient in feedback or assessment.[54] Students in the study by Henriksen and Ringsted[67] expressed scepticism about patients' knowledge[67] and expressed concerns about unstructured experiential learning in a context where patients had autonomy in both planning and delivering the teaching encounter.[67] In a different study, staff members expressed the concern that the impact of the patient experience might be reduced if the same patient was involved in the same programme too often.[66]

## Patients' views of the impact of their involvement
Patients described a strong sense of having a meaningful contribution and personal fulfilment, because they were teaching patient-centredness,[24 72] offering their body and authenticity, bolstering students' confidence,[72] fulfilling their responsibility to the broader community[24 32] and improving the healthcare system.[24 33 40]

On an individual level patients described material, professional, personal and emotional benefits. Material benefits included receiving tangible rewards such as gifts[43] and receiving a full medical check-up.[40] Patient educators with back pain involved in teaching medical students stated their participation improved the management of their own back pain, and improved confidence in voicing their needs in consultations with physicians.[47] Some patients felt that they received more time and attention from their healthcare professionals when they were teaching.[71]

Patients described professional growth and personal fulfilment from being involved in the selection process of students.[32] Hatem *et al* reported practical benefits for

patients including getting better at finding healthcare providers and increased knowledge of their medical condition.[43]

The drawbacks and risks associated with patient involvement in medical education included being confronted with stigmatising assumptions, vulnerability of self-disclosure and spontaneous question-answer exchanges. A patient educator teaching on the subject of HIV, for example, described the experience of being very frustrated with one man's lack of knowledge and ignorance about the disease. Patients also drew attention to the fact that unanticipated disease progression had an impact on their ability to teach. In some cases, this resulted in them pulling out of their commitment as teachers, an inevitable loss among patient educators.[43] Patients also described a sense of vulnerability to negative and non-appreciative reactions from students.[40 47] Initially, mentors were commonly anxious and unsure about whether what they shared was of benefit to students.[29] Half of the patients involved in the community-based intervention in a socioeconomic-deprived area expressed feelings of anxiety, apprehension or nervousness prior to the interview, although in all cases patients felt that this was normal.[52] In addition to the word 'vulnerable', patients employed terms like 'exposed', 'frightened', 'tired', 'stressed' and 'harrowing' to emphasise service users' feelings within the clinical education environment.[71] Some even described it as traumatic for mental health service users to repeatedly tell their stories.[71]

## Financial implications of patient involvement in medical education

Nineteen articles commented on any financial aspect of the interventions, ranging from reimbursement of patients' expenses, payments of honoraria, organisational costs or perceptions of cost. Economic and financial resources, however, have not been explored in a way that they can be systematically compared. Reported financial costs included $800 for a disability skills workshop,[63] £800 for a physical examination training[54] and £2640 for the overall Patient Partners programme.[47] Ten studies offered participating patients remuneration through honoraria between €8 per hour and £350 per day[43 53 54] or an unspecified amount.[28 35 39 48 50 57] Four studies offered reimbursement of patient expenses such as travel, phone or mail costs.[24 34 47 65] Some staff feared that cash patients needed to spend on refreshments or public transport would be a barrier for their participation.[73]

Medical educators suggested that patient involvement was a costly endeavour, both in financial investment as well as staff time.[71] Only one article commented on cost-effectiveness, noting that patient-led teaching is a cost-effective method compared with physician-led teaching, but did not provide an economic evaluation.[54] No paper provided a cost-effectiveness or cost-benefit analysis.

## Roles of patient organisations

The most cited interaction between patient organisations and medical faculties was the use of patient organisations and their networks for the recruitment of individual patients as community members or members of condition-specific support groups.[24 28 29 36 49 55 56 60 69] In the paper by Baral *et al*, representatives from rural communities and consumer groups were consulted by the medical school steering committee for the development of the Academy of Health Sciences curriculum.[45] Representatives of community-based patient advocacy and support organisations took part in the advisory group of the intervention in the study by Towle and Godolphin.[49] Not all of these representatives of patient organisations were patients themselves. They were described as brokers between two cultures of academia and community.

The University of Leeds worked with a dedicated internal patient group named 'The Patient Voice Group', consisting of lay people who use their experiences to inform their roles as teachers and researchers. This group was involved in formative and summative assessment. Additionally, a patient and public involvement manager who provided ongoing support was assigned within the school.[39]

Some medical educators made the explicit choice not to collaborate with patient organisations, due to a fear of working with politicised groups.[71] They did make a decision to include groups of patients to allow multiple voices to be heard in order to prevent criticisms of tokenism. Patients stated that participating in groups gave them support and companionship from their peers.[71]

Patient organisations wishing to engage in medical education may wish to consider some of the practical points as described in box 1 .

## Measures to ensure the sustainability of patient involvement

The key factors identified in sustaining patient involvement were the provision of adequate resource support, formal acknowledgement of the value of lay contributions and a clear faculty commitment to change following lay input.[26]

Institution-wide incorporation of social accountability or patient-centred education and medicine in the university's mission and vision statement or strategic plan was cited in several papers to ensure patient and community involvement. The authors emphasised that the resource intensity of a patient involvement programme requires the university to value its patient-centred underpinnings in order to be sustainable.[26 44 45 61] The incorporation of initiatives as ongoing modules in the curriculum achieved sustainable patient involvement rather than sporadic involvement.[53 63]

In the University of Leeds, a permanent patient voice group was incorporated in the institution.[39] The institution appointed a patient and public involvement manager to provide ongoing support. Some initiatives chose to work in partnership with existing institutions, implemented at a school-wide level[45] or focused on one

condition such as arthritis.[58] Gaver et al[55] identified the process of establishing commitment among volunteering organisations and families as a key challenge to the sustainability of patient involvement.[55]

**Box 1   Practical points for patient organisations wishing to engage in medical education**

Practical points for patient organisations
- ► Highlight potential benefits of involvement in medical education to members and respond to common motivations.
- ► Facilitate peer support networks among patients involved in medical education.
- ► Investigate whether teaching facilities in your area have existing frameworks for patient and public involvement, have made explicit commitments to patient and public involvement and/or have appointed officers specifically assigned to patient and public involvement that could provide you with a starting point for collaboration.
- ► Call for the creation of a patient committee or advisory group within a medical faculty to enable and safeguard structural input into the educational process.
- ► Determine which step of the educational process is best suited for your goals. You might choose to focus your efforts on the mission and vision of the school, learning objectives, educational strategies, teaching, assessment of learners or the evaluation of the course.
- ► Not every meaningful intervention has to be at an overall faculty level, they may also be on specific areas such as one learning objective or a disease-specific course.
- ► Showcase successful collaborations with universities and best practices to the wider patient and academic communities.

**Box 2   Spectrum of patient involvement in medical education by Towle et al[2]**

Levels of patient involvement in medical education as defined by Towle et al[2]
1. Patients as focus of a paper-based or electronic case.
2. Standardised or volunteer patients in clinical settings.
3. Patients sharing their experiences with students in a faculty-directed curriculum.
4. Patient teachers involved in teaching or evaluating students.
5. Patient teachers as equal partners in student education, evaluation and curriculum development beyond specific courses, to the curriculum as a whole.
6. Patients involved at the institutional level in addition to sustained involvement as patient teachers in education, evaluation and curriculum development.

Medical educators commented that if patient educators were paid and seen as an employee of the medical school, they might take on the role more seriously and become more reliable, as well as being seen as a respected part of the educational team.[66]

## DISCUSSION

This review systematically evaluated 49 primary empirical studies and was aimed at providing updated integrated evidence on the role and impact of the active involvement of patients in medical undergraduate education. The new body of empirical evidence shows the increasing range of learning objectives and educational settings in which patients play an active part in undergraduate medical education.

Our study found that patients described material, professional, personal and emotional benefits of participating in medical education. In addition to expected benefits, several authors mentioned policy mandates as rationale for initialising patient involvement programmes. Several studies however reported on the potential harms and negative experiences, such as fear of stigmatisation, tokenism or lacking structure of teaching session. Concerns related to patient involvement coming from students, faculty and patients themselves should remain closely monitored in a systematic manner and addressed appropriately.

Included papers described various types of roles for patients, but the vast majority of papers cited the role of a patient teacher, similarly to previous reviews.[1 17–19] More recent papers suggest that patients are increasingly involved in curriculum development. Most of these initiatives were incidental and were lacking institutional incorporation and longitudinal involvement.

The patients' roles identified in this review are largely in accordance with levels 3–6 of the spectrum proposed in the review by Towle et al[2] (box 2). We additionally identified new roles that could not be ascribed to one specific level on this spectrum. In one role, patients did take on roles as equal in curriculum development, but only to specific courses rather than the curriculum as a whole, falling between Towle's levels 4 and 5. In another role, patients were consulted in institution-level topics and curriculum development beyond specific courses, but rather than being equal partners, they were consulted in a faculty-driven initiative, displaying partial elements of Towle's levels 4, 5 and 6.[2]

The learning objectives identified in this review encompassed all but one of the CanMEDS roles for future physicians. This demonstrates that patient's involvement is continuing to gain a larger influence on a diverse range of aspects of the medical curriculum.

Measures to support sustainable patient involvement included longitudinal institutional incorporation, patient recruitment and/or training, resource support and clear commitment by faculty. The importance and advantages of patient involvement were highlighted by students, faculty and patients themselves; however, organisations must continue to consider, monitor and take steps to mitigate any potential harms to patients and students. Only few papers reported on the financial aspects related to patient involvement, which should be further investigated to help support feasibility.

An important limitation was the lack of common terminology in the existing literature, potentiating the risk of missing relevant articles, which has been previously reported as a limitation by other review authors.[2]

Our systematic review included only original literature from peer-reviewed journals. As many articles written by patients on their involvement in medical education may only be found in grey literature, including blog posts, conference statements and patient organisation newsletters, this review runs a risk of having missed important aspects of patient views on this topic. Additionally, only studies written in English were included, which may have led to bias in selected papers. The majority of included papers were from North America (n=23), Europe (n=17) and Australia and New Zealand (n=7).

The majority of included studies were qualitative (n=38), others were mixed methods (n=5) or qualitative (n=6). Only few of the included papers used control groups (n=7). In most cases, students and patients participated in interventions on a voluntary basis, which may limit the generalisability of findings to the wider population. Another important factor that may have introduced bias is that most studies were not (possible to be) anonymised, or were part of student assessments, which introduces a risk of responses being subject to social desirability bias.

Future research should focus on the long-term effects for patients, students and the healthcare system, especially on the subjects of patient-centredness and shared decision-making. This gap in research limits recommendations that can be made based on current literature. Additionally, no paper performed an economic evaluation of patient involvement, which may be a critical factor for decision makers in educational policy. Finally, more research is needed to update existing frameworks for patient involvement to the newly identified roles and needs patients have in medical education.

## CONCLUSION

It has been over 40 years since the first article on patient involvement in medical education was published. Today, both the medical education community and the patient community have joined together in the movement to promote patient-centredness. This systematic review provides knowledge and practical considerations that can aid curriculum developers who wish to sustainably incorporate active patient involvement in their institutions, and patient organisations wishing to engage in medical education.

**Acknowledgements** The authors thank Tessa Richards (BMJ Patient and Public Involvement lead) for her valuable comments and suggestions.

**Contributors** SWD and ED jointly developed the review protocol and search strategy. SWD, ED and MW jointly developed the data extraction sheet. The search strategy was critically reviewed by a hospital information specialist. Data collection was carried out by SWD and ED and extraction was carried out by SWD, ED and MW. The paper was written jointly by all three authors.

**Funding** The authors have not declared a specific grant for this research from any funding agency in the public, commercial or not-for-profit sectors.

**Competing interests** None declared.

**Patient consent for publication** Not required.

**Provenance and peer review** Not commissioned; externally peer reviewed.

**Data availability statement** All data relevant to the study are included in the article or uploaded as supplementary information. The authors confirm that the data supporting the findings of this study are available within its supplementary materials.

**ORCID iD**
Stijntje Willemijn Dijk http://orcid.org/0000-0003-2905-4128

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
