## [Reviewer comments · BMJ Open]

ARTICLE DETAILS

TITLE (PROVISIONAL)	The role of active patient involvement in undergraduate medical education: A systematic review
AUTHORS	Dijk, Stijntje; Duijzer, Edwin; Wienold, Matthias

VERSION 1 – REVIEW

REVIEWER	Morris Gordon University of central Lancashire I authored a BEME guide on this topic and STORIES statement, both of which I Cite in my review
REVIEW RETURNED	14-Feb-2020

GENERAL COMMENTS	This is a good piece and important, although with some work by the authors that are achievable, it will be of much higher relevance and quality. In terms of background, there was a recent publication by a team from BEME in medical teacher that covered the same topic. They focussed on the impact on the education of the learners, but not on the analysis from user perspective. As such, this is key to cite and links to methods. It isn't clear exactly where this review sits within this complexity. I have characterised my thoughts - any paper, any method, undergrad medical education and users, but am I right that the papers did not to actually provide a specific intervention or indeed assess how effective that ones in anyway? The narrative approach - I don't think that is the best choice for the paper. With the sorts of studies included, I would think a qualitative synthesis that has more scholarly output would be helpful. This would help explore how and why this involvement has impact to users and to learners. This is not a requirement for authors, but I think they should strongly consider the potential value a narrative synthesis misses. Also, PRISMA is ok, but if narrative, there are specific guidelines and frameworks to report these that should be followed and cited. Also, there is the STORIES statement for reporting educational reviews. I would love to see a revision
--

REVIEWER	Terese Stenfors Karolinska Institutet, Sweden
REVIEW RETURNED	18-Feb-2020

GENERAL COMMENTS	Thank you for providing me with this opportunity to read your work in an area of great importance. I found the manuscript well written and interesting. The methods section requires some further work, please describe your review methodology in more detail and specifically the analysis of the identified papers: Information is lacking on the qualitative synthesis you conducted and/if/why any templates were used for this (such as the CanMed framework). You have excluded a number of papers due to 'wrong article type', please exemplify. I also see no evidence of any quality assessment of the included studies? You mention in the discussion that the strength of the evidence is low, I suggest you develop this statement further and include suitable assessment tools for qualitative research. The result section is long and perhaps no so easy to follow, it may benefit from a clearer structure in subheadings or an illustration/summary of the structure. The discussion could be expanded to clearer match the objectives stated in the abstract. You may want to consider developing the supplementary table further, for example the data in the column for 'methods' could be synthesized so some clear types of data were identified. As some columns refer to the intervention at large and others to the research conducted to asses the intervention, the table is hard to follow, for example, are the outcome measures the learning outcomes of the intervention or the outcomes studies in the research? 'Research focus' can perhaps be split into research aim and intervention focus respectively and so forth. This is an important area of research and a well written paper. I look forward to following your work.
--

VERSION 1 – AUTHOR RESPONSE

Comments from Reviewer 1:

- Comment: In terms of background, there was a recent publication by a team from BEME in medical teacher that covered the same topic. They focused on the impact on the education of the learners, but not on the analysis from user perspective. As such, this is key to cite and links to methods. It isn't clear exactly where this review sits within this complexity. I have characterized my thoughts - any paper, any method, undergrad medical education and users, but am I right that the papers did not to actually provide a specific intervention or indeed assess how effective that ones in anyway?

Response: We were happy to refer to the newly published BEME guide and would like to congratulate the research team for the insightful and elaborative work that has been done. At the time of our initial submission, this paper had not yet been published. We have cited the paper in our revision. We agree that indeed a major difference between the two reviews is the main perspective, as our review is mainly written from the perspective of the patients opposed to the learner and educator's perspective. We have now explicitly added this notion in our discussion of existing reviews.

Our paper does not primarily address the effectiveness of specific interventions for student learning, but rather synthesizes any relevant information on a wide range of aspects of patient involvement as reported in the paper. These aspects include among other things, the type of patient involvement, how patients, students or faculty experience that involvement, and what the rationale behind the involvement was. When effectiveness was reported by included papers, we do report them, which you may mainly find in the section "Learning objectives pursued through active patient involvement".

• Comment: The narrative approach - I don't think that is the best choice for the paper. With the sorts of studies included, I would think a qualitative synthesis that has more scholarly output would be helpful. This would help explore how and why this involvement has impact to users and to learners. This is not a requirement for authors, but I think they should strongly consider the potential value a narrative synthesis misses.

Response: The reviewer raises a fair point when it comes to the method of synthesis most appropriate for this review of mostly but not exclusively qualitative research. A number of different methods have been proposed for the synthesis of qualitative findings, many based on approaches used in primary research. It is important to note that there are many different terms used to describe the various methods, some of which have been applied inconsistently. For our review we have deliberately chosen a broad definition and scope of patient involvement in undergraduate medical education, also resulting in a broad range of different methodology, ranging from RCTs and non-randomized interventions to grounded theory research and case studies. Considering the wide scope of aspects, we address, and the wide range of study designs we encounter, we think a pragmatic narrative synthesis is best suited for our review. We acknowledge this method of qualitative synthesis has its limitations, but we believe it does not inordinately restrict the practical implications and suggestions for future research of our work.

• Comment: Also, PRISMA is ok, but if narrative, there are specific guidelines and frameworks to report these that should be followed and cited. Also, there is the STORIES statement for reporting educational reviews.

Response: We would like to thank the reviewer for raising this point and providing an alternative suggestion for a statement. Our choice for using the PRISMA statement however has come from the guidelines as set forth by BMJ Open, the journal to which we have submitted our paper. We have also considered other frameworks such as the RAMESES and STORIES statement and considered each of the required reporting aspects, however felt that using one statement, as the preferred statement by the journal, would be preferable. If the reviewers and editors believed it to be essential to report a second statement such as RAMESES or STORIES, we would be willing to do so.

Comments from Reviewer 2:

• Comment: The methods section requires some further work, please describe your review methodology in more detail and specifically the analysis of the identified papers: Information is lacking on the qualitative synthesis you conducted and/if/why any templates were used for this (such as the CanMed framework). You have excluded a number of papers due to 'wrong article type', please exemplify. I also see no evidence of any quality assessment of the included studies? You mention in the discussion that the strength of the evidence is low, I suggest you develop this statement further and include suitable assessment tools for qualitative research.

Response: We have expanded on our methods section by elaborating on our data collection methods, the items on our data extraction sheet, the framework used for the extraction of intended learning outcomes, the excluded article types and the quality assessment of included studies. We have added the quality assessment using the MMAT as a supplementary file, and we've edited the results section to better reflect our findings.

• Comment: The result section is long and perhaps no so easy to follow, it may benefit from a clearer structure in subheadings or an illustration/summary of the structure.

Response: In an attempt to make the result section easier to follow, we added a summary of the structure at the start of the section. A change we have made to support the reviewers in the revision (but not related to the final article), is changing the layout of our manuscript to better reflect the layout of BMJ Open, so that the reviewers may better see how the different subheadings will be distinguishable if the article is accepted. We hope the adjustments have made it easier for the

reviewers to follow this section and would welcome any additional remarks to further aid readers to follow.

• Comment: The discussion could be expanded to clearer match the objectives stated in the abstract.
Response: We would like to thank the reviewer for this comment. We agree and believe that the abstract better fitted a previous version of the discussion than our most recent one. We have changed the order and structure of the discussion section, further expanded this section, and improved the alignment between abstract and discussion.

• Comment: You may want to consider developing the supplementary table further, for example, the data in the column for 'methods' could be synthesized so some clear types of data were identified. As some columns refer to the intervention at large and others to the research conducted to assess the intervention, the table is hard to follow, for example, are the outcome measures the learning outcomes of the intervention or the outcomes studies in the research? 'Research focus' can perhaps be split into research aim and intervention focus respectively and so forth.
Response: Thank you for pointing this out, we agree with your suggestions. We have adjusted our supplementary table by creating new columns and splitting information related to the research question, study design, data collection method and the setting of the intervention. We have reduced the number of words where possible, and changed the layout of the table, to hopefully make it easier to follow.

VERSION 2 – REVIEW

REVIEWER	Morris Gordon University of Central Lancashire School of Medicine, WELFARE, PROFESSIONALISM, TRANSITION AND CAREERS
REVIEW RETURNED	04-May-2020
GENERAL COMMENTS	Thanks for considering Whilst I still feel not using another form of synthesis is underselling the potential of the piece, it is up to the editor to consider. I have no further comments